

# Burden in caregivers of children with congenital Zika syndrome in Pernambuco, Brazil: analysis and application of the Zarit burden interview scale

Jerônimo Faustino Rego Filho[1], Claudia Sena[2] and Rubens Wajnsztejn[1]

[1] Centro Universitário Da Faculdade De Medicina Do Abc Paulista, São Paulo, Brazil
[2] Universidade de Pernambuco, Recife, Brazil

## ABSTRACT

With the increase in cases of microcephaly caused by the Zika virus, the demand for special care and a better quality of life for the child and caregiver increased proportionally.

**Objective**. This study aimed to analyze the burden on caregivers of children with congenital Zika syndrome associated with viral infections in the state of Pernambuco, Brazil using the Zarit Burden interview scale.

**Method**. A quantitative study was conducted at the Oswaldo Cruz University Hospital, Recife City, State of Pernambuco, Brazil. By convenience sampling, 56 mothers, two grandmothers, and two caregivers were enrolled, all are female. Data were collected from July 2019 to January 2020. In the analysis, the percentage frequencies were calculated. The normality was identified using the Kolmogorov–Smirnov test, and participant profiles were compared using Student's $t$-test and analysis of variance. In descriptive statistics, quantitative variables are described by the median and interquartile range and categorical variables by proportions using the Chi-square test.

**Results**. In the comparative analysis, all factors evaluated were significant, except for the "gestational period in which the disease occurred" ($p < 0.111$). The significance of differences in all activities was evaluated. In the mean comparison test, only the factor "has a job" was significant ($p < 0.043$). When comparing the average of caregivers' responses to the categories of the Zarit burden interview scale, the highest level of burden was regarding the feeling that the child is dependent on the caregiver (3.62 points). Caregiver burden was classified as absence, light, moderate, and high.

**Conclusion**. The consequences of contracting the Zika virus in the first trimester of pregnancy, lack of paid work, financial scarcity, full-time dedication to the child, and lack of time for themselves increase the burden on caregivers. Thus, caregivers have mild burden.

Corresponding author
Jerônimo Faustino Rego Filho,
jeronimofausto@yahoo.com.br

## INTRODUCTION

The Zika virus is a neurotropic stranded ribonucleic acid arbovirus of the genus *Flavivirus*. It is found in rhesus macaques in the Zika Forest, Uganda, in 1947. *Flavivirus* infection results in a rash infection of epidemiological importance because of its binding to several types of neuronal cells, which may be associated with neurological changes in newborns (*Felipe et al., 2021*; *Khongwichit et al., 2021*). In 2016, the World Health Organization confirmed the causal association between congenital microcephaly and the Zika virus, observing the development of congenital neurological malformations associated with viral infection in pregnant women, highlighting the congenital Zika syndrome. Since then, a public health emergency of international importance has emerged (*Khongwichit et al., 2021*; *Bosaipo et al., 2019*; *de Sá et al., 2020*).

In March 2016, the Ministry of Health in Brazil created a surveillance protocol in response to the increase in microcephaly cases in the country. From week 45 of 2015 to week 45 of 2020, 3,563 cases of the congenital Zika syndrome were confirmed in Brazil. Most cases were concentrated in the Northeast region (61.9%). In Pernambuco, 13.1% were recorded in this period, of which 78% affected infants, and 74 fetal deaths were confirmed, of which 45 occurred in 2016. In 2021, Brazil recorded 5,710 probable cases of congenital Zika syndrome. The Northeast region accounted for 4,252 cases, and Pernambuco added 610 cases to this total (*Ministry of Health BR, 2020*; *Ministry of Health , BR*).

Microcephaly is defined by the deformation of the skull size, inappropriate for age and sex at birth. It is a complex and multifactorial etiological anomaly and was diagnosed by measuring the head circumference 24 h after birth. In the first week of life, microcephaly is noted when the head circumference is less than two or more standard deviations in relation to age, sex, or gestational age. The World Health Organization adopted a head circumference of ≤31.5 cm for girls and 31.9 cm for boys (*Torquato et al., 2020*; *Vargas et al., 2016*).

Alterations in the circulatory, digestive, auditory, musculoskeletal, ocular systems and, mainly, nervous systems are associated with the Zika virus, resulting in some motor and cognitive alterations. The most common are intellectual deficit, brain paralysis, epileptic seizures, autism, and attention-deficit hyperactivity disorder (*Department, 2015*; *Azevedo, Freire & Moura, 2021*).

With the increase in cases of microcephaly associated with the Zika virus, the demand for special care and a better quality of life for the child–caregiver binomial increased proportionally. Family members of children with microcephaly face considerable challenges such as reduced income, marital and family breakdown, uncertainties about the child's progression, and the opportunities they will have in the future (*de Sá et al., 2020*). Among the various concerns and difficulties faced by caregivers, a relevant factor is that it is the child's accessibility, which, depending on the degree of need, causes mental and physical burden to caregivers (*Lima, Cardoso & Costae Silva, 2016*).

In this sense, some scales and instruments measure the general burden of caregivers, such as interviews and competency questionnaires. One of the most used in the assessment of caregiver burden is the Zarit burden interview scale, which is a revision of the original

version. It consists of 29 items covering aspects of psychological and physical health, economic resources, work, social relationships and the relationship with the care receiver.

The Zarit burden interview scale is self-administered and allows assessing the objective and subjective burden of informal caregivers, collecting information related to their health, social life, personal situation, pecuniary, emotion, and relationship consisting of 22 items on a Likert scale. Each item is scored quantitative/qualitative as follows: never = (0); rarely = (1); sometimes = (2); many times = (3); always = (4). In the version used (0 to 4), an overall score ranging from 0 to 88 is obtained. The higher the score, the greater the perceived overload (*Dry, 2010*; *Scazufca, 2002*).

Society has changed in terms of the demographic, epidemiologic, and nutritional aspects of the population. Consequently, there were demographic advances, urbanization processes, and agglomerations, causing the emergence of several epidemics. Despite this, technological advances in health have led to a reduction in infant mortality and survival of newborns with congenital alterations. Thus, there is an increasing need for attention and care, reinforcing the reorganization of health parameters to maintain the quality of life of children and their caregivers.

Therefore, this study aimed to analyze the burden on the caregivers of children with congenital Zika syndrome in the state of Pernambuco, Brazil, by analyzing the results of the Zarit burden interview scale. Therefore, understanding the process from the emergence of the virus to the care burden is important.

## MATERIALS AND METHODS

This cross-sectional, exploratory-descriptive study with a quantitative approach was conducted at the Oswaldo Cruz University Hospital, University of Pernambuco (HUOC/UPE), located in Recife City, State of Pernambuco, Brazil. It is a reference institution in the care of children with microcephaly caused by the Zika virus. The study population consisted of 60 caregivers of children with microcephaly caused by the Zika virus, including 56 mothers, two grandmothers, and two caregivers.

As a selection criterion, caregivers of children diagnosed with microcephaly caused by the Zika virus and treated at the aforementioned health center were included. Caregivers who refused to participate in the study, even after signing the free and informed consent term, caregivers who had been admitted to the unit for <6 months, and visually impaired caregivers were excluded. Convenience sampling was used because only a small number of eligible patients were identified. The study variables were sex, kinship with the child, age, marital status, gestational period in which the signs and symptoms of the Zika virus disease occurred, family income, profession, time spent caring for the child, time dedicated to the child, and activities performed as a caregiver.

The instruments for data collection were a self-administered sociodemographic questionnaire and the Zarit burden interview scale to analyze the burden of caregivers of children with microcephaly. This care questionnaire was developed by a group of researchers headed by Steven Zarit, a professor at the University of Pennsylvania, and it was translated and validated for Brazil by *Scazufca (2002)*.

Data collection took place in the pre-appointment (waiting room). All caregivers who met the inclusion criteria were invited to participate in the study. They received information about the research content, in which all doubts were clarified, and they were assured of the confidentiality of both their identities and data. Data were collected from July 2019 to January 2020.

For data analysis, a database was built in the electronic spreadsheet using Microsoft Excel (which was exported to the SPSS Statistics version 18; SPSS Inc., Chicago, NY, USA).

To characterize the personal profile of caregivers and the practice of caring for children with microcephaly, the percentage frequencies were calculated, and the respective frequency distributions were constructed. The Chi-square test was used to compare the proportions of categorical variables.

For the evaluation of the Zarit score items, the prevalence of caregivers' responses to the questionnaire items was obtained. In addition, the total Zarit score was obtained for the burden of caregivers of children with microcephaly. Score normality was assessed using the Kolmogorov–Smirnov test. If the score showed a normal distribution, the caregivers' profile was compared using the two-tailed Student's $t$-test for variables with two categories, and a one-way analysis of variance test was used to analyze variables with three or more categories. Descriptive statistics was also used, in which quantitative variables were described as median and interquartile range and categorical variables as proportions using the Chi-square test to verify whether a relationship exist between the care burden of caregivers.

Finally, care burden was classified; each item was scored 0 to 4 on the Likert scale: 0, never; 1, rarely; 2, sometimes; 3, often; and 4, always. The sum of the scores was considered absence (<21 points), light (21–40 points), moderate (41–60 points), and high (61–88 points) burden. A significance level of 5% was considered.

The study was approved by the Ethics Committee for Research on Human Beings (CEP) of the University Center of the Faculty of Medicine of ABC Paulista under Opinion No. 3,373,788 under CAAE: 13840419.6.0000.0082. All ethical and legal aspects were respected as recommended by Resolutions 466/12 and 510/2016 of the National Health Council.

## RESULTS

Table 1 shows the distribution of the sociodemographic profile of the caregivers of children with microcephaly caused by the Zika virus. Of the total of 60 female caregivers (100.0%), 93.4% were the mothers of the affected children. Of these mothers, 47% were between 31 and 40 years old, 65.0% were single, and 31.7% were in the first trimester of pregnancy when they showed signs and symptoms of Zika disease. As regards income, 96.7% received 1 Minimum Wage (1 MW), and 98.3% do not have paid work. As shown in Table 1, the proportion of all factors compared was significant ($p < 0.05$), indicating that the profile described is common among the evaluated caregivers, except for the factor "gestational period that the disease occurred" ($p < 0.111$), which indicates that the disease appears with similar prevalence in the three gestational trimesters.

Table 2 shows the comparison of the mean and standard deviation of the Zarit score for caregiver burden according to the sociodemographic profile. The mean burden score

**Table 1  Distribution of the sociodemographic profile of caregivers of children with microcephaly.** Recife, PE, Brazil, 2021 ($n = 60$).

| Evaluated factor | $n$ | % | $p$-value[a] |
|---|---|---|---|
| Sex | | | |
| Female | 60 | 100.00 | 0 |
| Kinship | | | |
| Mother | 56 | 93.4 | |
| Grandmother | 2 | 3.3 | <0.001 |
| Caregiver | 2 | 3.3 | |
| Age | | | |
| <20 years | 1 | 1.0 | |
| 21–30 years | 25 | 42.0 | <0.001 |
| 31–40 years | 28 | 47.0 | |
| >41 years | 6 | 10.0 | |
| Marital status | | | |
| Married | 19 | 31.7 | |
| Single | 39 | 65.0 | <0.001 |
| Divorced | 2 | 3.3 | |
| Gestational period in which the disease occurred | | | |
| 1st trimester | 19 | 31.7 | |
| 2nd trimester | 15 | 25.0 | <0.111 |
| 3rd trimester | 7 | 11.6 | |
| No information | 19 | 31.7 | |
| Family Income | | | |
| 1 Minimum Wage[*] | 58 | 96.7 | |
| 2 Minimum Wage[*] | 2 | 3.3 | <0.001 |
| Paid employment | | | |
| Yes | 1 | 1.7 | |
| No | 59 | 98.3 | <0.001 |

Notes.
[a]$p$-value of the Chi-square test for the proportion comparison.
[*]Current minimum wage = BRL 1,212.00, Brazil, 2022.

was higher in caregivers who were related to the child (mean = 40.0), aged <20 years (mean = 49.0), divorced (average = 40.5), diagnosed with the disease in the first trimester of pregnancy (mean = 37.1), eared 1 MW (average = 32.8), did not receive treatment (average = 33.6), and had paid work (average = 59.0). In the mean comparison test, only the factor "has a job" was significant ($p < 0.043$), indicating that having paid work outside the home, in addition to caring for the child, significantly increases the burden score.

As shown in Table 3, all activities evaluated were significant ($p < 0.05$), indicating that the number of caregivers who practice such activities to care for children with microcephaly is statistically relevant.

Table 4 shows the distribution of responses in relation to the items of the Zarit burden interview scale. The categories most denied by caregivers were as follows: feel stressed in taking care of the child and their other responsibilities with the family and work (41.7%), feel ashamed of the child's behavior (98.3%), feel irritated when the child is around

**Table 2  Mean and standard deviation of the Zarit score for caregiver burden according to the personal profile.** Recife, PE, Brazil, 2021 ($n = 60$).

| Evaluated factor | Overload level | | p-value |
|---|---|---|---|
| | Average | DP[a] | |
| Kinship | | | |
| Mother | 33.2 | 13.0 | |
| Grandmother | 15.5 | 2.1 | 0.127[b] |
| Caregiver | 40.0 | 11.3 | |
| Age | | | |
| <20 years | 49.0 | 0 | 0.332[b] |
| 21–30 years | 29.8 | 12.4 | |
| 31–40 years | 34.2 | 12.4 | 0.332[b] |
| >41 years | 35.8 | 18.6 | |
| Marital status | | | |
| Married | 31.9 | 15.2 | |
| Single | 32.9 | 12.4 | 0.684[b] |
| Divorced | 40.5 | 7.8 | |
| Gestational period in which the disease occurred | | | |
| 1st trimester | 37.1 | 15.3 | |
| 2nd trimester | 36.5 | 12.3 | 0.085[b] |
| 3rd trimester | 27.0 | 8.2 | |
| No information | 28.5 | 11.0 | |
| Family income | | | |
| 1 Minimum Wage[*] | 32.8 | 13.2 | 0.930[c] |
| 2 Minimum Wage[*] | 32.0 | 17.0 | |
| Health treatment | | | |
| Yes | 30.0 | 11.3 | 0.386[c] |
| No | 33.6 | 13.6 | |
| Paid employment | | | |
| Yes | 59.0 | 0 | 0.043[c] |
| No | 32.4 | 12.8 | |

Notes.
[a] standard deviation.
[b] p-value of the ANOVA test.
[c] p-value of Student's t-test.
[*] Current minimum wage = BRL 1,212.00, Brazil, 2022.

(88.4%), feel that the child negatively affects their relationships with other family members or friends (75.0%), feel tense when the child is around (85.0%), feel that their health was affected because of their involvement with the child (51.7%), feel that they do not have as much privacy as they would like because of the child (48.3%), feel that their social life has been harmed due to having to take care of the child (65.0%), does not feel comfortable having visitors at home because of the child (85.0%), feel that they will be unable to take care of the child for much longer (76.6%), feel that he/she has lost control of his/her life since the child's illness (68.4%), would simply like to let someone else take care of the child

**Table 3** Distribution of activities performed by caregivers of children with microcephaly. Recife, PE, Brazil, 2021 ($n = 60$).

| Analyzed activities | Perform the activity | | * *p*-value[a] |
|---|---|---|---|
| | **Yes** | **No** | |
| Alimentation | 56 (93.3%) | 4 (6.7%) | <0.001 |
| Medication | 55 (91.7%) | 5 (8.3%) | <0.001 |
| Shower | 56 (93.3%) | 4 (6.7%) | <0.001 |
| Clothing | 56 (93.3%) | 4 (6.7%) | <0.001 |
| Transportation | 56 (93.3%) | 4 (6.7%) | <0.001 |

**Notes.**
[a] *p*-value of the Chi-square test for the proportion comparison.

(58.4%), feel in doubt about what to do for the child (53.4%), and feel overwhelmed by caring for the child (46.7%).

Still, as shown in Table 5, most caregivers stated the following categories that occur frequently: feel that the child asks for more help than he/she needs (41.7%); feel that because of the time they spend with the child, they do not have enough time for themselves (43.4%); fear for the child's future (38.3%); feel that the child depends on you (90.0%); feel that the child expects you to take care of them as if you were the only person they can depend on (53.3%); feel that they do not have enough money to take care of the child, in addition to their other expenses (68.4%); feel that they should be doing more for the child (48.3%); and feel that they could take better care of the child (36.7%).

Table 6 shows normality and data values using the Kolmogorov–Smirnov non parametric test. Normal parameters: mean (32.8167), standard deviation (13.12197). More extreme differences: absolute (.089), positive (.089), negative (−.086). Statistical test (.089) and bilateral significance (.200).

The findings show that 71.7% of the caregivers have a mild level of burden, followed by 26.7% with a moderate burden and 1.6% with a severe burden, and the absence of burden was 0%.

## DISCUSSION AND CONCLUSION

Of the 60 female caregivers, mothers (mean age, 31–40 years), single, were in the first trimester of pregnancy when they showed signs and symptoms of congenital Zika syndrome. Almost all the factors evaluated in the proportion comparison test were significant. However, when comparing the mean and standard deviation of the Zarit score for caregiver burden, according to the sociodemographic profile, only the factor "has a job" was significant ($p < 0.043$), indicating that having paid work outside the home, in addition to caring for the child, significantly increases the burden score. The study showed that most caregivers have a mild level of burden (71.7%), followed by those with a moderate level (26.7%).

In this study, all caregivers of children with microcephaly caused by the Zika virus were women, with the predominance of mothers as caregivers. In this sense, mothers who care for a child with a congenital malformation tended to be responsible for the
**Table 4** **Distribution of the responses to the items of the Zarit burden interview scale, category that never occurs.** Recife, PE, Brazil, 2021 (n = 60).

| Zarit scale | Never | Rarely | A few times | Often | Always | Average |
|---|---|---|---|---|---|---|
| 1. Stressed about taking care of the child? | 25 41.7% | 7 11.7% | 14 23.3 | 3 5.0% | 11 18.3% | 1.47 |
| 2. Embarrassed by the child's behavior? | 59 98.3% | 0 0.0% | 1 1.7% | 0 0.0% | 0 0.0% | 0.03 |
| 3. Angry when the child is around? | 53 88.4% | 1 1.7% | 2 3.3% | 2 3.3% | 2 3.3% | 0.32 |
| 4. Does the child negatively affect your relationships? | 45 75.0% | 4 6.7% | 5 8.3% | 0 0.0% | 6 10.0% | 0.63 |
| 5. Are you tense when the child is around? | 51 85.0% | 1 1,7% | 6 10.0% | 0 0.0% | 2 3.3% | 0.35 |
| 6. Has your health been affected because of the child? | 31 51.7% | 7 11.7% | 10 16.6% | 1 1.7% | 11 18.3% | 1.23 |
| 7. Don't have as much privacy as you would like? | 29 48.3% | 7 11.7% | 11 18.3% | 3 5.0% | 10 16.7% | 1.30 |
| 8. Has your social life been affected? | 39 65.0% | 4 6.6% | 7 11.7% | 3 5.0% | 7 11.7% | 0.92 |
| 9. Don't feel comfortable having visitors? | 51 85.0% | 1 1.7% | 2 3.3% | 2 3.3% | 4 6.7% | 0.45 |
| 10. Will you be unable to care for the much longer? | 46 76.8% | 4 6.6% | 4 6.6% | 5 8.3% | 1 1.7% | 0.52 |
| 11. Have you lost control of your life? | 41 68.4% | 3 5.0% | 8 13.3% | 3 5.0% | 5 8.3% | 0.80 |
| 12. Would you like to simply let someone else take care? | 35 58.4% | 2 3.3% | 9 15.0% | 5 8.3% | 9 15.0% | 1.18 |
| 13. Do you feel in doubt about what to do? | 32 53.4% | 3 5% | 17 28.3% | 3 5.0% | 5 8.3% | 1.10 |
| 14. Do you feel like you should be doing more? | 16 26.7% | 4 6.7% | 8 13.3% | 3 5.0% | 29 48.3% | 2.42 |
| 15. You feel overwhelmed? | 28 46.7% | 3 5.0% | 6 10.0% | 7 11.0% | 16 26.6% | 1.67 |

integral care of the child. In addition, she tends to adapt to the new reality and redirect her concepts and prejudices about children with disability; a similar pattern was found in the studies by *Delfino et al. (2021)*; *Nam & Park (2017)*. Arguably, full dedication can result in work burden, compromise their quality of life, and cause or worsen health problems for caregivers.

Regarding sociodemographic data, most caregivers are single, which shows the constant maternal presence in the care of the child. Dedication to the child with special needs, often without the help of the child's father, can cause excessive tiredness, exhaustion, and frustration for caregiver-mothers. The absence of a partner in doing care activities and concerns about the health of the child with microcephaly reverberate in mothers through the lack of time for their personal care. Corroborating the results of *Pinto et al. (2016)* the data show that divorced caregivers had a higher level of burden.
**Table 5  Distribution of responses to the items of the Zarit burden interview scale, a category that always occurs frequently.** Recife, PE, Brazil, 2021 ($n = 60$).

| Zarit scale | Never | Rarely | A few times | Often | Often | Average |
|---|---|---|---|---|---|---|
| 1. Does the child ask for more help than he or she needs? | 19 31.7% | 3 5.0% | 8 13.3% | 5 8.3% | 25 41.7% | 2.23 |
| 2. Because of the time you spend with the child, don't you have enough time for yourself? | 9 15.0% | 3 5.0% | 14 23.3% | 8 13.3% | 26 43.4% | 2.65 |
| 3. Fear for the child's future? | 16 26.7% | 3 5.0% | 16 26.7% | 2 3.3% | 23 38.4% | 2.22 |
| 4. Does the child depend on you? | 5 8.3% | 1 1.7% | 0 0.0% | 0 0.0% | 54 90.0% | 3.62 |
| 5. Does the child expect you to take care of him/her as if you were the only person he/she can depend on? | 13 21.7% | 6 10.0% | 6 10.0% | 3 5.0% | 32 53.3% | 2.58 |
| 6. Don't have enough money to take care of the child in addition to your other expenses? | 8 13.3% | 3 5.0% | 5 8.3% | 3 5.0% | 41 68.4% | 3.10 |
| 7. Do you feel that you could take better care of the child? | 21 35.5% | 5 8.3% | 7 11.7% | 5 8.3% | 22 36.7% | 2.03 |

**Table 6  Normality of the Zarit scale score results using the Kolmogov-Smirnov test.** Recife, PE, Brazil, 2021 ($n = 60$).

| | | SUM DE ZARIT |
|---|---|---|
| N | | 60 |
| Normal parameters[a,b] | Mean | 32,8167 |
| | Standard deviation | 13,12197 |
| More extreme differences | Absolute | ,089 |
| | Positive | ,089 |
| | Negative | -,086 |
| Test statistic | | ,089 |
| significance sig. bilateral | | ,200[c,d] |

**Notes.**
NPAR TESTS
/K-S(NORMAL)=SUM_ZARIT.
/MISSING ANALAYSIS
TEST OF A SAMPLE OF Kolmogorov–Smirnov
[a] The test distribution is normal.
[b] Calculated from the data.
[c] Correction of significance of Liliiefors.
[d] This is a lower bound of true significance.

As caregivers were not employed, possibly due to the need to provide comprehensive care to the child, the family income is low. By compromising relationships between family members, low family income can demonstrate the urgency of creating measures, strategies, and public policies that can support such families in a state of physical and emotional vulnerability. Therefore, the results of the present study strengthen the findings of *Félix & Fárias (2018)*; *Padilha et al. (2017)* who discussed the importance of the social security law and public access to the health system for children. The recognition of the child with microcephaly as a disability, according to Ordinance No. 58 of June 3, 2016, presents the

need to assist these families. Caregivers receive regular benefit that helps them with the expenses of the children (*Government of Brazil, 2016*).

Children with microcephaly demand time and full attention from caregivers, increasing the workload. These children have high-risk levels because of alterations in the neuropsychomotor development, such as prematurity and abnormal growth of the head circumference. If there is abnormal neurogenesis, synaptogenesis, or neuronal migration in the first and second trimesters of pregnancy, brain growth and development are hindered, and congenital infections, neurological changes, cerebral palsy, attention-deficit hyperactivity disorder, behavioral disorders, and vision, hearing, motor coordination, and learning problems occur. These conditions can even interfere with the actual care delivery. The finding corroborates those of *Pires et al. (2019)*, *Ministry of Health BR (2016)*, *Ferreira et al. (2018)* and *Sá et al. (2017)*.

The light burden (71.7%) identified in caregivers appears to be related to marital status, age, and degree of kinship with the child. As regards the degree of limitations, motor, cognitive, visual, and auditory alterations make it difficult for caregivers to provide care, as children progressively become more dependent. This finding is similar to that found by *Ribeiro et al. (2017)* on microcephaly caused by the Zika virus, as it causes serious damage and changes in neurological development, with a high potential for overloading caregivers. *Torquato et al. (2020)* and *Moore et al. (2017)* reported similar results.

As for the division of tasks, in which spouses or other family members share concerns and tasks, there are fewer risks to the physical and mental health of those who play the role of the primary caregiver. These data were verified by *Reis et al. (2015)*. The vast majority of caregivers directly perform all the daily tasks of the child, such as bathing, feeding, transporting, etc. This reinforces children's dependence and caregivers' burden.

As the study limitations, this is a descriptive study; thus, more in-depth conclusions regarding the burden of caregivers of children with microcephaly caused by the Zika virus as well as other issues such as the correlation of some variables cannot be inferred. The use of convenience sampling and closed questions of the instrument did not allow considerations from the caregivers' point of view in relation to the care burden of the Zarit burden interview scale. A qualitative study could provide better answers to these questions.

Therefore, a multicenter study is recommended to obtain a greater number of participants and deepen the data. Since the research was carried out in a single care service for patients with congenital microcephaly syndrome caused by the zika virus. Therefore, the generalized results of caregivers being all female participants, mostly single, unemployed, do not represent the entire population of caregivers in Pernambuco, Brazil. Thus, institutional support, an adequate place for consultations and procedures, proper disclosure, and promotion of scientific research on the subject are essential. Government investments in public policies and the psychosocial well-being of caregivers are also important, allowing for development in caring not only for the child but also for the caregiver.

Microcephaly is a disabling chronic condition; therefore, health professionals and managers should understand the real care requirement of a family member with this
condition and the consequences in the life of the caregiver. This can improve caregivers' quality of life, reflecting better care for children with microcephaly.

The absence of primary care programs for caregivers promotes factors that generate burden. Investing in prevention support can promote the quality of life of caregivers and children, minimizing the burden of care.

This study presents relevant points because it used a standardized and validated instrument in Brazil and has already been applied in studies that portray the importance of caregiver satisfaction and quality. Furthermore, these findings could be a good indicator of the health of caregivers of children with microcephaly caused by the Zika virus and may contribute to the planning of public actions aimed at this population.

The proportion comparison test showed that microcephaly appears with similar prevalence in the three gestational trimesters. When genetic deformations occur more frequently during pregnancy, caring for a child with a neurological disability can generate weariness, as it makes tasks increasingly difficult to perform because of time constraints. The mean comparison test was significant only for the factor ''has a job,'' indicating that having paid work outside the home significantly increases the burden level, in addition to the exhaustive care for the child, lack of family support, other house chores, lack of infrastructures, and poor financial condition.

When comparing the average of caregivers' answers to the items on the Zarit burden interview scale, caregivers reported a higher burden level on these items: feel that the child is dependent on you, feel that he/she does not have enough money to take care of the child in addition to other expenses, and feel that spending time with the child reduces the time for oneself.

Microcephaly is a disabling chronic condition; thus, health professionals and managers should understand the real care requirement of a family member with this condition, considering its consequences on the life of the caregiver. This can improve caregivers' quality of life, reflecting better care for children with microcephaly.

Contracting the Zika virus in the first trimester of pregnancy, resulting in the impossibility of having paid work outside the home, not having enough money for expenses, total dependence on child care, and lack of time for personal care are factors that increase the care burden on caregivers. Overall, the survey score of the participants was classified as light burden.

## ACKNOWLEDGEMENTS

The preparation of this thesis would not have been possible without the collaboration, encouragement, and commitment of several people. I want to express my sincere thanks to everyone.

### Funding
The authors received no funding for this work.

## Competing Interests
The authors declare there are no competing interests.

## Author Contributions

- Jerônimo Faustino Rego Filho conceived and designed the experiments, performed the experiments, analyzed the data, prepared figures and/or tables, authored or reviewed drafts of the article, and approved the final draft.
- Claudia Sena conceived and designed the experiments, performed the experiments, analyzed the data, prepared figures and/or tables, authored or reviewed drafts of the article, and approved the final draft.
- Rubens Wajnsztejn conceived and designed the experiments, performed the experiments, analyzed the data, prepared figures and/or tables, authored or reviewed drafts of the article, and approved the final draft.

## Human Ethics

The following information was supplied relating to ethical approvals (i.e., approving body and any reference numbers):

The Ethics Committee for Research on Human Beings (CEP) of the University Center of the Faculdade de Medicina do ABC Paulista approved the study (13840419.6.0000.0082).

## Data Availability

The raw data are available in the Supplemental Files.

## Supplemental Information

Supplemental information for this article can be found online at http://dx.doi.org/10.7717/peerj.14807#supplemental-information.

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
