# Peer review of "Burden in caregivers of children with congenital Zika syndrome in Pernambuco, Brazil: analysis and application of the Zarit burden interview scale"

_PeerJ, doi:10.7717/peerj.14807_

## Round 0.1 · original submission · Minor Revisions

I have now received the reviewers' comments on your manuscript. They have suggested some minor revisions to your manuscript. Therefore, I invite you to respond to the reviewers' comments and revise your manuscript.

Reviewer 1 ·

Basic reporting

1. Since all your participants were females, the authors may want to emphasize that in the title and/or abstract.

Experimental design

1. In the Introduction section, “One of the most used in the assessment of caregiver burden is the Zarit burden interview scale, which is a revision of the original version.” What is the original version? Could the authors also provide more details about the “Zarit burden interview scale”?
2. More details are needed for Zarit score items/system as well.

Validity of the findings

1. In the Materials and Methods section, “If the score showed a normal distribution, the caregivers. profile was compared…” What about the scores that didn’t show normality? Did the authors apply any transformation or other analysis approaches?
2. The study only involved female participants, which may cause bias.

Reviewer 2 ·

Basic reporting

The paper is well-written. Enough background and literature references are provided. The structure is very clear. The only concern is that the authors did not mention whether the data could be shared.

Experimental design

The research topic falls into the scope of the journal. The research question is well-defined and the methods are described clearly.

Validity of the findings

The conclusions are appropriately stated and can answer the original research question. But I have some concerns about how their results can support their findings. See more details in the additional comments.

Additional comments

1. Have the pvalues reported in the tables been adjusted for multiple comparisons or the raw pavlaues? If not, please adjust for multiple comparisons.
2. The authors claim that normality of the data was identified using the Kolmogorov3Smirnov test. Please report the results.
3. Are the statistical test of each variable (Kinship, age, etc) mentioned in table 2 performed separately? If so, why not fit a multivariate regression model at one time?
4. The authors finally observed the light burden (71.7%) identified in caregivers of children with congenital Zika syndrome. What is the number for caregivers of healthy children? Is there any significant difference? I hope the authors can compare the results with the caregivers of healthy children to make their findings more persuasive.
5. In table 2, the first value is 0.127, instead of 0,127.

---

## Round 0.2 · Minor Revisions

I have now received the reviewers' comments on your manuscript. They have suggested some minor revisions to your manuscript. Therefore, I invite you to respond to the reviewers' comments and revise your manuscript.

Reviewer 1 ·

Basic reporting

No comment.

Experimental design

No comment.

Validity of the findings

The authors should provide explicit discussion about study limitations due to all female participants of whom mostly single, unemployed, since this group may not represent the caregiver population in Pernambuco, Brazil.

Reviewer 2 ·

Basic reporting

The authors addressed my concerns. No additional comments at this time.

Experimental design

The authors addressed my concerns. No additional comments at this time.

Validity of the findings

The authors addressed my concerns. No additional comments at this time.

---

## Round 0.3 · accepted · Accept

In my opinion this manuscript has been revised with attention to the reviewers' comments and can now be published.